# Retrospective analysis of real-world data to evaluate actionability of a comprehensive molecular profiling panel in solid tumor tissue samples (REALM study)

**Karen Leroy** [1,2]*, **Clarisse Audigier Valette**[3], **Jérôme Alexandre**[1,4], **Lise Boussemart**[5], **Jean Chiesa**[6], **Clotilde Deldycke**[7], **Carlos Gomez-Rocca**[8], **Antoine Hollebecque**[9], **Jacqueline Lehmann-Che**[10,11], **Antoinette Lemoine**[12], **Sandrine Mansard**[13], **Jacques Medioni**[14], **Isabelle Monnet**[15], **Samia Mourah**[10,16], **Thomas Pierret** [17], **Dominique Spaëth**[18], **Alexandre Civet**[19☯], **Sandrine Galoin**[20☯], **Antoine Italiano**[21☯]

1 Université Paris Cité, Sorbonne Université, Inserm, Centre de Recherche des Cordeliers, Paris, France, 2 Département de Médecine Génomique des Tumeurs et Cancers, Service de Biochimie, AP-HP, Hôpital Européen Georges Pompidou, Paris, France, 3 Pôle Médecine à Orientation Oncologique, Hôpital Sainte Musse, Toulon, France, 4 Service d'Oncologie, AP-HP, Hôpital Cochin, Paris, France, 5 Service de Dermatologie, CHU de Nantes—Hôtel Dieu, Nantes, France, 6 UF de Cytogénétique et Génétique Médicale, Hôpital Universitaire Carémeau, Nîmes, France, 7 Pôle Régional de Cancérologie—CHU de Poitiers, Poitiers, France, 8 Oncologie Médicale, IUCT Oncopole, Toulouse, France, 9 DITEP, Gustave Roussy, Villejuif, France, 10 Université Paris Cité, INSERM U976, Immunologie Humaine, Pathophysiologie, Immunothérapie (HIPI), Paris, France, 11 UF Oncologie Moléculaire, Hôpital Saint-Louis, AP-HP, Paris, France, 12 Biochimie et Oncogénétique–Inserm UMRS 1193, Hôpital Paul Brousse, AP-HP, Paris, France, 13 Service de Dermatologie, CHU-Estaing, Clermont-Ferrand, France, 14 Centre d'Essais Précoces en Cancérologie, Hôpital Européen Georges Pompidou, Paris, France, 15 Service de Pneumologie, Hôpital Intercommunal de Créteil, Créteil, France, 16 Service de Génomique des Tumeurs et Pharmacologie, Hôpital Saint-Louis, AP-HP, Paris, France, 17 Onco-Pneumologie, Hospices Civils de Lyon, Lyon, France, 18 Centre d'Oncologie de Gentilly, Institut Interrégional de Cancérologie, Nancy, France, 19 Centre de Données Médicales, Roche S.A.S, Boulogne-Billancourt, France, 20 Affaires Médicales, Roche S.A.S, Boulogne-Billancourt, France, 21 Unité d'études de Phases Précoces, Institut Bergonié, Bordeaux, France

☯ These authors contributed equally to this work.
* karen.leroy@aphp.fr

## Abstract

### Introduction

Considering the growing interest in matched cancer treatment, our aim was to evaluate the ability of a comprehensive genomic profiling (CGP) assay to propose at least one targeted therapy given an identified genomic alteration or signature (actionability), and to collect the treatment modifications based on the CGP test results in clinical practise for solid tumors.

### Methods

This retrospective, multicentre French study was conducted among 25 centres that participated in a free of charge program between 2017 and 2019 for a tissue CGP test. Data were collected on the patient, disease, tumor genomic profile, treatment suggested in the report (related to the genomic profile results) and subsequent therapeutic decisions according to the physician's declaration.

**Data Availability Statement:** All relevant data are within the paper and its Supporting Information files.

**Funding:** "This study was funded by Roche S.A.S. The funders have provided the FICDx tests, they supervised the study design, funded the CRO who managed data collection, and participated to the data analysis, decision to publish, and preparation of the manuscript, as detailed in the author contribution section".

**Competing interests:** "I have read the journal's policy and the authors of this manuscript have the following competing interests: Karen Leroy: Roche- board, conference fees, scientific collaboration, scientific meeting fees; AstraZeneca, BMS- board, conference fees, scientific meeting fees; Lilly, Janssen- board; Amgen, MSD, GSK- scientific meeting fees; Nanostring- conference fees, scientific collaboration. Clarisse Audigier Valette: AstraZeneca, Boehringer Ingelheim- Financial Interests, Personal, Principal Investigator, Advisory Role; BMS, Lilly, Novartis, Pfizer-Financial Interests, Personal, Invited Speaker, Advisory Role; Abb Vie, GlaxoSmithKline, Janssen, MSD, Roche, Sanofi, Takeda-Financial Interests, Personal, Advisory Role. Antoine Italiano: Roche- Financial Interests, Personal, Advisory Board, Research Grant Roche. This does not alter our adherence to PLOS ONE policies on sharing data and materials."

## Results

Among the 416 patients, most had lung cancer (35.6%), followed by biliary tract cancer (11.5%) or rare cancers (11.1%); 75% had a metastatic disease. The actionability was 75.0% (95% CI [70.6%-78.9%]) for all patients, 85.1% and 78.4%, respectively in lung cancer and metastatic patients. After exclusion of clinical trial suggestions, the actionability decreased to 62.3% (95% CI [57.5%-66.8%]). Treatment modification based on the test results was observed in 17.3% of the patients and was more frequent in metastatic disease (OR = 2.73, 95% CI [1.31–5.71], p = 0.007). The main reasons for no treatment modification were poor general condition (33.2%) and stable disease or remission (30.2%). The genomic-directed treatment changes were performed mostly during the first six months after the CGP test, and interestingly a substantial part was observed from six to 24 months after the genomic profiling.

## Conclusion

This French study provides information on the real-life actionability of a CGP test based on tissue samples, and trends to confirm its utility in clinical practice across the course of the disease, in particularly for patients with lung cancer and/or advanced disease.

## Introduction

Since the identification of clinically relevant cancer genes, the current therapeutic trend has been to identify genomic alterations that can be targeted with matched available drugs, either approved or assessed in clinical trials, as this could translate into patient benefit [1–3]. Considering the increasing number and variety of clinically relevant genomic alterations occurring across a large number of cancer-related genes, large genomic profiling is currently proposed [4–8]. Multigene sequencing avoids performing numerous independent techniques and allows large analysis, while sparing tissue samples and time to choose the most appropriate treatment.

FoundationOne® [9], from Foundation Medicine, Inc. (FMI), Boston, USA, is a send-out tissue-based comprehensive genomic profiling test for solid tumors, CLIA (Clinical Laboratory Improvement Amendment) certified, fulfilling the requirements of the European Directive 98/79 EC for *in vitro* diagnostic medical devices (IVD), and CE marked [9]. An updated version of this test, FDA approved as a companion diagnostic solution in 2019, is available as FoundationOne® CDx (F1CDx) [10]. Using formalin-fixed and paraffin-embedded (FPPE) tumor tissue specimens, this next generation sequencing (NGS) test allows the 4 genomic alteration classes identification (mutations, insertions/deletions (indels), amplifications and rearrangements) in 324 genes known to be altered in solid tumors, as well as to assess microsatellite instability (MSI), tumor mutational burden (TMB) and genomic loss of heterozygoty (gLOH) in the updated version for ovarian cancer [9, 10]. The tumor genomic profile is detailed in the F1CDx report, and somatic alteration(s)/signature(s) are associated with available targeted therapie(s) and immunotherapie(s) approved in patient tumor type or in other tumor types, and/or currently assessed in clinical trials. FoundationOne® has been used to profile over 100,000 patients for more than 100 types of solid tumors [11].

The current challenge of CGP tests is their clinical relevance in oncology practice, with discussions aiming to define the most suitable situations in which CGP should be prescribed in a real-life setting: which genes (prognostic/predictive) for which tumors? What are the time

points of the patient's journey? (diagnosis and/or each progression? first line or second line or more for advanced/metastatic disease) [12]. These questions led to the first European Society for Medical Oncology (ESMO) recommendations in 2020 for routine use of NGS, notably for advanced non-squamous non-small-cell lung cancer (NSCLC), prostate cancer, ovarian cancer, and cholangiocarcinoma [13]. In this context, it is of particular interest to provide information on the actionability and clinical utility of CGP tests, which were both assessed in this observational study.

## Methods

### Study design

REALM is a retrospective, non-interventional, multicentre French study, conducted between 2019–2020 among oncologists, pathologists, and biologists who participated in a free of charge (FoC) program for tissue F1CDx test between April 2017 and September 2019. This study was proposed to the centres where at least 10 FoC tests were performed. In accordance with the French laws (Loi Jardé), an ethical committee approval was not required and not sought for because the REALM study was qualified as a research involving only secondary use of health data. In such studies, no written consent is requested, but all patients need to be informed about the use of their data. The REALM protocol was compliant with the methodology of reference (MR-004) [14] edited by the French National Commission for the protection of private data and rights. All patients included in this study were informed that their test results and associated medical information could be used for scientific research and gave initially a written consent to perform a F1CDx test between April 2017 and September 2019. Prior to the beginning of the study, all eligible alive patients received a written information form explaining that a study using their health data was put in place and that they had the right to object if they did not want their data to be used in the study. The eCRF could only be filled if the clinician certified that the patient did not oppose to this study.

### Participants

A total of 25 centres (16 university hospitals, 6 cancer centres, 3 general hospitals) agreed to participate in the study. Eligible patients were adults (≥18 years) with a solid tumor, who were informed as described above and did not oppose to the use of their data.

### Data collection

Retrospective data were collected at three points: 1) before F1CDx request (patient demographic data, smoking status, disease and biomarker history, and therapeutic classes of previous cancer treatment), 2) at the time of F1CDx test [ECOG performance status, comorbidities, cancer staging and disease progression if applicable, current treatment line, date and type of tumor tissue specimen and test result (genomic profile and matched proposed therapies, approved, off-label or experimental)], and 3) after F1CDx report availability in the center (changes in patient treatment, according to the test results or not, and reasons for no changes, date of treatment start and stop, date of last news). The follow-up period was defined as the time between the F1CDx test and the last data collected for this study (last contact, death, or loss to follow-up). All collected data were fully anonymous and as declared by the physician who filled the electronic form.

### Definition of an actionable case

A patient's tumor profile was considered actionable following CGP test results when at least one treatment was proposed according to an identified genomic alteration(s)/signature(s) in the F1CDx report. This treatment could be an approved drug, or an association of drugs, in patient's tumor type or in another tumor type, or investigational drug(s) in a clinical trial.

### Statistical analysis

A descriptive analysis was performed. Study parameters were assessed using the mean and standard deviation or median and interquartile range for continuous variables, and proportions for categorical variables. Two-sided tests with type I error $\alpha = 0.05$ were applied when relevant and were considered significant at $p < 0.05$. Statistics analysis was performed using SAS® software (SAS Institute, Cary, NC, USA), version 9.4. Analysis was performed according to the type of solid tumor at initial diagnosis [patients with lung cancer, patients with rare tumors (cancer incidence <6/100,000/year in the general population) [15] and patients with other tumors], and according to cancer staging (patients with and without metastasis at the time the F1CDx was received, respectively for all patients and in the lung group). The cumulative incidence of the actionability in oncology practice was described from the reception of the F1CDx report, using the Kalbfleisch and Prentice method and considering patient death as a competitive event in order to limit bias estimation [16]. Factors influencing genomic-directed treatment were examined for patient characteristics (age, sex, ECOG performance status, number of comorbidities, and smoking status), cancer staging at the time the F1CDx test was performed, cancer treatments (before and at the time the F1CDx was realized), and results from the F1CDx (number of alterations identified and TMB signature evaluation), using logistic regressions. Univariate analyses were performed to select (p ≤0.25, and missing data <25%) the explanatory variables to be included in the multivariate model, and those that were statistically significant (p <0.05) were retained for the final model.

## Results

### Patient and disease characteristics

Of the 511 included patients, 416 (81.4%) were included in the analysis. Among the 95 excluded patients, 57 patients had no F1CDx report available, 23 patients had a so-called "qualified" (incomplete) report, 8 patients had F1CDx test performed in the context of a clinical trial, and 7 patients had a F1Liquid CDx test performed. Among the 416 evaluable patients, 148 (35.6%) had lung cancer, 46 (11.1%) had rare tumors, and 222 (53.4%) had other tumors (Fig 1). Most patients (75.0%) had metastatic disease at the time of the F1CDx test. Among 148 patients with lung cancer, 136 (91.9%) had metastatic disease. Patient characteristics, at the time physicians requested the F1CDx test, are detailed for all evaluable patients in the Table 1, limited to patients with metastatic disease in S1 Table and in most frequent cancer types outside lung cancer in S1 File. Overall, the mean age of the patients was 58.9 ± 12.9 years, and 46.6% were older than 60 years. Most patients were in good general condition (ECOG performance status<2: 80.9%). Patients with lung cancer had biomarker testing prior to the F1CDx test more often than the patients with other cancers (in 59.5% of the cases *versus* 13.0% for rare tumors, and 25.7% for other tumors). The testing method (multiple choice allowed) used for lung cancer cases was reported for 88 patients: immunohistochemistry 37.5%, fluorescent in situ hybridization 10.2%, targeted-PCR 8% and sequencing 89.8% without information regarding the specific genes that had been tested. Overall, F1CDx tests were performed 14.5 months

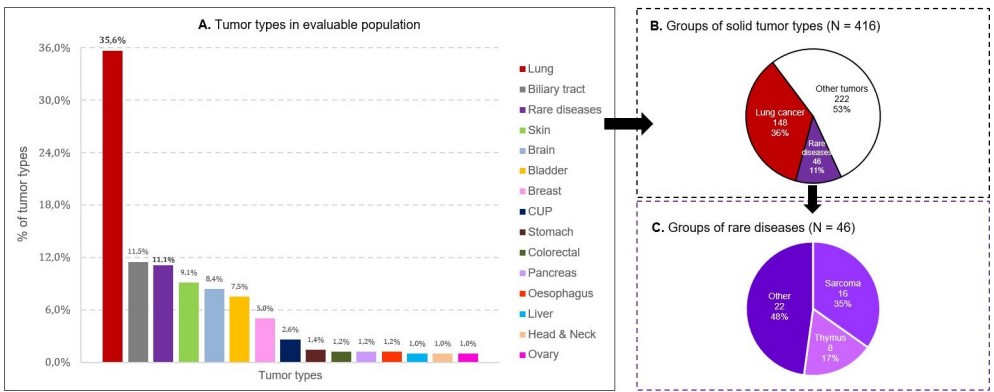

**Fig 1. Solid tumor types at F1CDx request time in the evaluable population (N = 416).** Cancer histology details: Lung (148): non-small cell carcinoma (125 adenocarcinoma, 12 squamous, 4 large cell, 2 sarcomatoid, 1 neuroendocrine, 3 other) and 1 small cell carcinoma; Sarcoma (16): angiosarcoma (1), chordoma (1) conventional chondrosarcoma (1), clear cell sarcoma (1), fibrosarcoma (1), hemangiopericytoma (1), leiomyosarcoma (2), meningeal melanocytoma (2), mesenchymal chondrosarcoma (2) stromal sarcoma of the sex cords (1), sarcoma (3, no details); Thymus (8): thymoma (2), carcinoma (6); Other (in Rare group, 22): adenoid cystic tumor (1), adrenocortical cancer (1), aggressive digital papillary adenocarcinoma (1), appendiceal carcinoma (1), digestive neuro-endocrine carcinoma (1), clear cell carcinoma of the endometrium (1), esthesioneuroblastoma (1), intestinal-type adenocarcinoma of the ethmoide (1), germinal tumor (1), lymphoepithelial carcinoma of the larynx (1), epithelioid mesothelioma (2), pituitary carcinoma (1), pleural mesothelioma (1), malignant teratoma (1), paraganglioma (1), anal squamous cell carcinoma (2), ocular melanoma (1), ovarian clear cell carcinoma (2), undifferentiated neuroendocrine tumor of unknown primary (1).

in median after initial diagnosis; this time was shorter for patients with lung cancer compared to other patients (7.8 months *versus* 27.1 and 14.9 for rare and other tumors, respectively).

## Results of F1CDx tests

Table 2 presents the type and number of alterations identified, in the evaluable population and according to cancer type. Point mutations were mostly detected in *TP53* (in 45.9% of patients), *KRAS* (15.6%), and *TERT* (10.6%) (S2 Table). The different genes and types of alterations according to the gene families/pathways altered across all samples are shown in S2 File.

All patients with lung cancer had at least one genomic alteration (median number: 5). Single nucleotide variant (SNV) or indels (short insertions/deletions) were mostly detected in the following genes: *TP53 (73%)*, *KRAS (30.4%)*, *CDKN2A (25.7%)*, and *EGFR (*11.5%); most frequent amplification concerned *MYC* (12.2%). Gene rearrangements were identified in 5/148 lung tumors (four of which involved *ALK*, *ROS1* or *RET*). Patients with rare tumours had at least one genomic alteration in 88.6% of the cases, and the median number of alterations was two. Rearrangements involving *ESWR1* gene were detected in two fibrosarcomas and one sarcoma (not specified); two mesenchymal chondrosarcomas showed the characteristic *HEY1*::*NCA2* fusion. In the "Other tumors group" four rearrangements involving *BRAF* gene were detected (three melanoma: *BRAF*::*CNNM2*, *BRAF*::*ERC1*, *BRAF*::*MAD1L1* and one pancreatic adenocarcinoma *BRAF*::*SND1*), but also one *ETV6*::*NTRK3* (colorectal cancer) and one *FGFR3*::*TACC3* (urothelial cancer).

The median TMB for the 416 evaluable patients was 11 mut/Mb, with the majority of low (≤5 mut/Mb, 45.6%) or intermediate (≥5 and ≤20, 40.6%) TMB and 13.9% of patients with a high TMB (≥20). The majority of patients with lung cancer (55%) had an intermediate TMB, and 22.9% had a high TMB. Considering a cut-off value of ≥ 10 mut/Mb (TMB associated to Pembrolizumab FDA approval in solid tumors, unresectable or metastatic, after progression

**Table 1. Characteristics of patients at F1CDx report in the evaluable population.**

| | Lung cancer | Rare tumors[a] | Other tumors[b] | Total |
|---|---|---|---|---|
| | **N = 148** | **N = 46** | **N = 222** | **N = 416** |
| **Demographics** | N = 148 | N = 46 | N = 222 | N = 416 |
| Age (years), mean±SD | 62.1 ± 10.2 | 51.9 ± 14.9 | 58.3 ± 13.4 | 58.9 ± 12.9 |
| Age ≥60 years | 62 (41.9) | 26 (56.5) | 106 (47.7) | 194 (46.6) |
| Male sex | 84 (56.8) | 27 (58.7) | 107 (48.2) | 218 (52.4) |
| **ECOG performance** | N = 141 | N = 44 | N = 207 | N = 392 |
| 0 | 36 (25.5) | 13 (29.5) | 62 (30.0) | 111 (28.3) |
| 1 | 79 (56.0) | 19 (43.2) | 108 (52.2) | 206 (52.6) |
| 2 | 16 (11.3) | 9 (20.5) | 30 (14.5) | 55 (14.0) |
| >2 | 10 (7.1) | 3 (6.8) | 7 (3.4) | 20 (5.1) |
| **At least one comorbidity*** | N = 142 | N = 43 | N = 192 | N = 377 |
| | 72 (50.7) | 16 (37.2) | 107 (55.7) | 195 (51.7) |
| **Metastatic disease** | N = 148 | N = 46 | N = 222 | N = 416 |
| | 136 (91.9) | 39 (84.8) | 137 (61.7) | 312 (75.0) |
| **Smoking status (at diagnosis)** | N = 147 | N = 21 | N = 222 | N = 142 |
| Former smoker | 63 (42.9) | 5 (23.8) | 38 (26.8) | 106 (34.2) |
| Smoker | 60 (40.8) | 4 (19.0) | 18 (12.7%) | 82 (26.5%) |
| Non-smoker ever | 24 (16.3) | 12 (57.1) | 86 (60.6) | 122 (39.4) |
| **Time from diagnosis to F1CDx test** | N = 148 | N = 46 | N = 222 | N = 416 |
| Months, Median (range) | 7.8 (0.6,125.2) | 27.1 (1.1,161.9) | 14.9 (0.6,327.6) | 14.5 (0.6,327.6) |
| **At least one genetic test performed prior to F1CDx test** | N = 148 | N = 46 | N = 222 | N = 416 |
| | 88 (59.5) | 6 (13.0) | 57 (25.7) | 151 (36.3) |

SD: standard variation

*Comorbidity: arterial hypertension, cardiovascular disease, dermatological disease, gastrointestinal disease, genital tract disease, infections, metabolic disease, neuromuscular disease, neuropsychiatric disease, ophthalmic disease, osteoarticular disease, other respiratory diseases, type I or II diabetes)

NOTE. Data presented as number (%) unless indicated otherwise

[a] Sarcoma (n = 16), thymic carcinoma (n = 8), other rare tumors (n = 22)

[b] Biliary tract (n = 48), skin (n = 38), brain (n = 35), bladder (n = 31), breast (n = 21), stomach (n = 6), colorectal (n = 5), pancreas (n = 5), esophagus (n = 5), liver (n = 4), mouth (n = 4), ovary (n = 4), hail intestine (n = 3), uterine (n = 1), uterine cervix (n = 1), unknown primary (n = 11)

following prior treatment and without satisfactory alternative treatment options), 58.6% of lung tumors, 2.2% of rare tumors, and 24.6% of other cancer types had an elevated TMB [17].

## Actionability

The median duration of patient follow-up was 8.6 months (interquartile range, [IQR]: 3.3–16.4, min-max range: 0–29.7). In the evaluable population of 416 patients, F1CDx test actionability was 75.0%, including drugs approved in the cancer type (40.7%), in another cancer type (42.3%), or only available in trials (17.0%). Around 17% of the evaluable population received a proposed genomic-directed treatment. Considering only the 312 patients with actionable alteration(s), 23% received a genomic-directed treatment. For the 72 patients who received a therapy based on F1CDx report, the main new treatments were targeted therapy (74.0% of the cases), and immunotherapy (21.9%) (S3 Table). The genomic-directed treatment could be standard of care in the indication (n = 23), investigational drugs within clinical trials (n = 20) or off label use of an available molecule (n = 29). The main reasons for not using the test results to guide treatment were poor general condition of the patient (in 23.0% of the cases), stable

**Table 2. Results of the F1CDx test in the evaluable population.**

| | Lung cancer | Rare tumors[a] | Other tumors[b] | Total |
|---|---|---|---|---|
| | *N = 148* | *N = 46* | *N = 222* | *N = 416* |
| **Genomic alterations** | N = 148 | N = 39 | N = 220 | N = 407 |
| Any genomic alterations identified | 148 (100.0) | 39 (88.6) | 220 (100.0) | 407 (98.8) |
| ]1–5] | 69 (46.6) | 26 (66.7) | 87 (39.5) | 182 (44.7) |
| ]5–10] | 63 (42.6) | 4 (10.3) | 47 (21.4) | 114 (28.0) |
| >10 | 6 (4.1) | 1 (2.6) | 13 (5.9) | 20 (4.9) |
| **Type of genomic alterations identified** | *N = 148* | *N = 39* | *N = 220* | *N = 407* |
| **Mutation or single-nucleotide variation** | | | | |
| Identification | 146 (98.6) | 36 (92.3) | 189 (85.9) | 371 (91.2) |
| Mean number ± SD | 3.6 ± 2.0 | 1.8 ± 0.8 | 2.8 ± 2.4 | 3.0 ± 2.2 |
| Median | 3.0 | 2.0 | 2.0 | 3.0 |
| Q1-Q3 | [2.0, 5.0] | [1.0, 2.0] | [1.0, 4.0] | [1.0, 4.0] |
| Range | (1.0,13.0) | (1.0,4.0) | (1.0,19.0) | (1.0,19.0) |
| **Insertion/deletion (indels)** | | | | |
| Identification | 51 (34.5) | 12 (30.8) | 71 (32.3) | 134 (32.9) |
| Mean number ± SD | 1.3 ± 0.6 | 1.3 ± 0.9 | 1.5 ± 0.7 | 1.4 ± 0.7 |
| Median | 1.0 | 1.0 | 1.0 | 1.0 |
| Q1-Q3 | [1.0, 2.0] | [1.0, 1.0] | [1.0, 2.0] | [1.0, 2.0] |
| Range | (1.0,3.0) | (1.0,4.0) | (1.0,4.0) | (1.0,4.0) |
| **Amplification or copy number variation** | | | | |
| Identification | 69 (46.6) | 11 (28.2) | 86 (39.1) | 166 (40.8) |
| Mean number ± SD | 2.7 ± 1.8 | 2.6 ± 2.3 | 2.9 ± 2.0 | 2.8 ± 1.9 |
| Median | 2.0 | 2.0 | 2.0 | 2.0 |
| Q1-Q3 | [1.0, 3.0] | [1.0, 3.0] | [1.0, 4.0] | [1.0, 4.0] |
| Range | (1.0,11.0) | (1.0,8.0) | (1.0,9.0) | (1.0,11.0) |
| **Rearrangement** | | | | |
| Identification | 7 (4.7) | 5 (12.8) | 11 (5.0) | 23 (5.7) |
| **Microsatellite status** | *N = 140* | *N = 44* | *N = 177* | *N = 361* |
| Stable | 139 (99.3) | 44 (100.0) | 173 (97.7) | 356 (98.6) |
| High | 1 (0.7) | 0 | 4 (2.3) | 5 (1.4) |
| **Tumor mutation burden** (mutations/ megabase) | *N = 140* | *N = 45* | *N = 175* | *N = 360* |
| Mean ± SD | 14.5 ± 12.6 | 3.9 ± 2.4 | 10.0 ± 17.3 | 11.0 ± 14.8 |
| ≤ 5 | 31 (22.1) | 34 (75.6) | 99 (56.6) | 164 (45.6) |
| [6–19] | 77 (55.0) | 11 (24.4%) | 58 (33.1) | 146 (40.6) |
| ≥ 20 | 32 (22.9) | 0 | 18 (10.3) | 50 (13.9) |

SD, standard variation

NOTE. Data presented as number (%) unless indicated otherwise

[a] Sarcoma (n = 16), thymic carcinoma (n = 8), other rare tumors (n = 22)

[b] Biliary tract (n = 48), skin (n = 38), brain (n = 35), bladder (n = 31), breast (n = 21), stomach (n = 6), colorectal (n = 5), pancreas (n = 5), esophagus (n = 5), liver (n = 4), mouth (n = 4), ovary (n = 4), hail intestine (n = 3), uterine (n = 1), uterine cervix (n = 1), unknown primary (n = 11)

disease or remission (20.9%), or no ongoing clinical trials available in France or at the participant site (10.7%) (Fig 2).

Actionability was higher in metastatic patients than in non-metastatic patients (78.5% *versus* 64.4%), including drugs approved in the cancer type (36.7%), in another cancer type (43.7%), or only available in trials (19.6%). Around 20% of patients with a metastatic disease

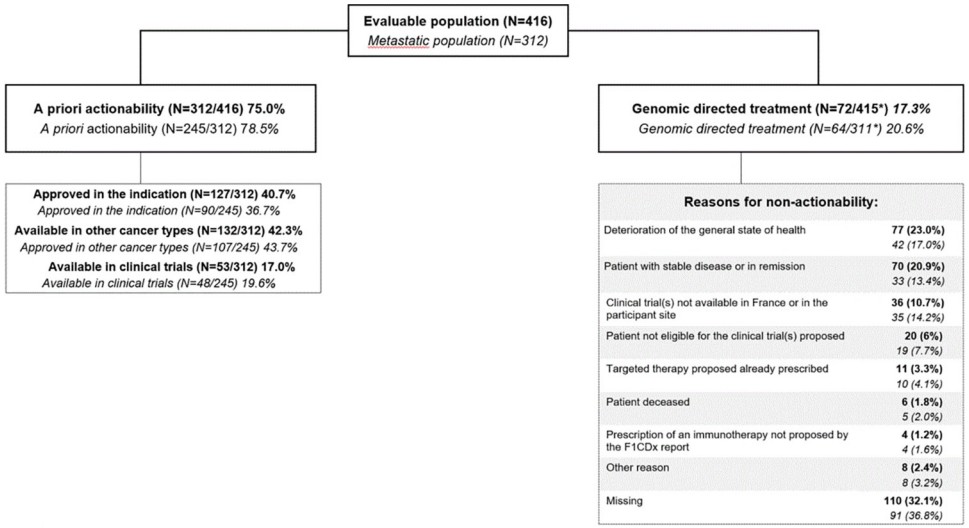

**Fig 2. Actionability of F1CDx test.** Actionability is defined as the ability of the F1CDx test to propose at least one treatment given an identified alteration(s) or signature(s). Data are shown for all evaluable patients (bold) and limited to metastatic patients at the time of F1CDx test (italics). CI, confidence interval. *One patient had no data provided regarding actionability in a real clinical setting.

received a proposed genomic-directed treatment (Fig 2). In metastatic patients, genomic-directed change was similar regardless of the line of treatment (19.0% for no treatment, 22.0% if 1st line, 19.7% if 2nd line, and 20.7% for at least 3rd line). Multivariate analysis showed that metastatic disease at the time of the F1CDx test was significantly associated with prescription of a matched therapy of the CGP report (OR = 2.74, 95% CI [1.31–5.71], p = 0.007) (Fig 3). In the majority of patients who received a genomic-directed therapy, it was initiated during the first 6 months after the tests, and interestingly a substantial part is observed from 6 to 24 months after the test: 13.8% at Month 6 (95% IC [10.5%-17.5%]), 17.7% at Month 12 ([13.9%-21.9%]), 20.0% at Month 18 ([15.9%-24.6%]), and 22.6% at Month 24 (95% IC [17.7%-27.9%]) (Fig 4).

In patients with lung cancer, the actionability was higher than that in other patients (85.1% *versus* 47.8% for rare tumors and 73.9% for other tumors), but a greater decrease was observed in lung cancer when investigational drug(s) were excluded from proposed matched (62.8% versus 41.3% and 66.2%, respectively). In addition, less patients received a proposed genomic-directed treatment in lung cancer (14.2% *versus* 23.9% and 18.6%, respectively). The main reason for not performing proposed matched therapy for patients with lung cancer were the lack of available or adapted clinical trials (34.6%), followed by the poor general condition of the patient (26.0%), and no disease progression (25.0%). For patients with lung cancer, three parameters were significantly associated with the genomic-directed treatment: ECOG performance status at the time of the F1CDx test (0 *versus* ≥1; OR = 3.12, 95% CI [1.02–9.62], p = 0.048), smoking status at initial diagnosis (non-smoker ever *versus* smoker: OR = 15.38, 95% CI [2.81–83.33], p = 0.002; non-smoker ever *versus* former smoker: OR = 1.65, 95% CI [0.50–5.49], p = 0.415), and clinical centers type (OR = 5.1, 95% CI [1.3–19.7], p = 0.018).

## Discussion

In the context of current discussions on the right moment to perform CGP test in oncology practice and for the right patient, this real-world retrospective French study aimed to assess the actionability associated with F1CDx on tissue samples for solid tumors [18]. In the

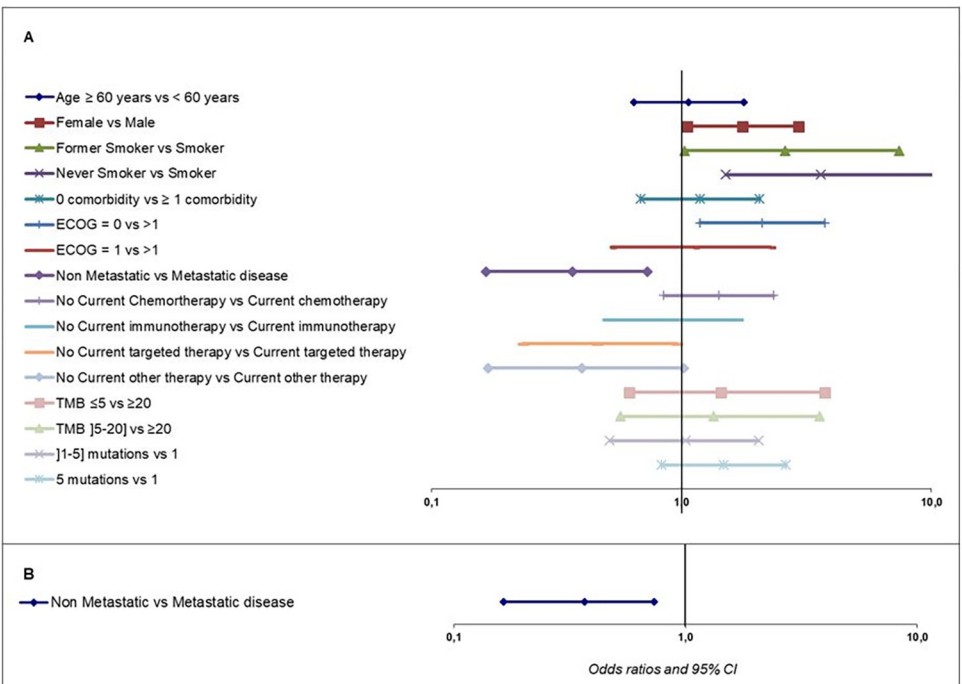

**Fig 3.** Univariate analysis (A) and Multivariate analysis (B) to identify potential factor influencing test-informed (genomic-directed) treatment of F1CDx test. CI, confidence interval; ECOG, Eastern Cooperative Oncology Group; TMB, tumor mutation burden.

REALM study, the most frequent mutations were observed in *TP53* and *KRAS* like in two others F1CDx retrospective cohort studies [19, 20], respectively Japanese in a pan-tumor population, and Finnish in lung cancer only (between 2004 and 2017). Even if patients with any solid

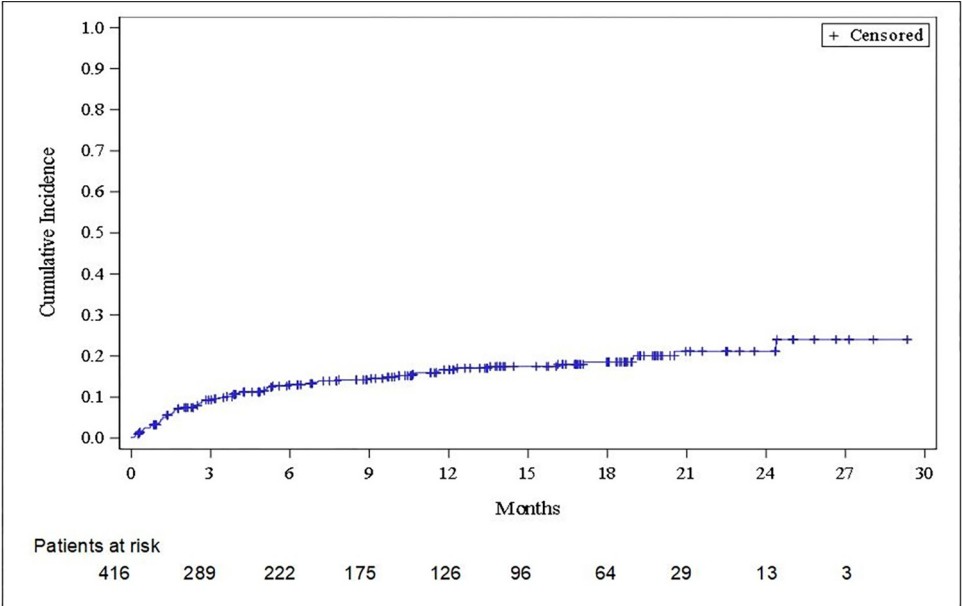

**Fig 4. Cumulative incidence of test-informed (genomic-directed) treatment, from the time when F1CDx results were available (Kalbfleisch and Prentice curve).** Patient death considered as competing event.

tumor were eligible in the REALM study, most of the tumor samples tested between 2017 and 2019 came from lung and/or advanced/metastatic tumors. The frequency of alterations in *KRAS* and *EGFR* in lung tumors was very similar to that observed in a 1-year nationwide program of the French Cooperative Thoracic Intergroup, indicating there was no major bias in the recruitment of these patients [21].

The overall F1CDx actionability was at 75.0%. This rate was among the highest from previously reported for other NGS [3, 4, 20, 22–25] but lower than two recently published Japanese cohort studies [19, 26], using F1CDx between mid-2018 and mid-2019. Differences could be explained by the study design (real-world *versus* randomized studies), the heterogeneity of assessments in terms of target patient populations (type of cancer, disease stage, ongoing line of cancer treatments, patient performance status, etc.), NGS techniques and gene panel contents, and the year of study conducted in view of the increasing number of matched drugs and immunotherapies. For example, the actionability increased over time in French studies involving patients with advanced cancer: it was 40% in the SHIVA trial [27] conducted between 2012 and 2014, 49% according to the MOSCATO-01 trial [3] conducted between 2011 and 2016, and 52% in the ProfiLER-01 trial [5] conducted between 2013 and 2017. In addition, the actionability should be better described according to validated actionability scales such as OncoKb [28] or ESCAT [29] to compare the performances of different genomic panels and testing strategies. Unfortunately, considering the timing of the REALM study, insufficient information on gene alterations was collected to re-analyse actionability on this basis.

The F1CDx actionability is higher in lung and/or advanced/metastatic tumors than in rare tumors and/or non-metastatic patients [30]. The highest actionability was observed in patients with lung cancer, which is in line with the high number of biomarkers identified and needed now for therapeutic decision. These findings are consistent with recent European [13] and French regional recommendations [31] for the routine use of NGS for lung cancer. Excluding F1CDx proposals for matched drugs assessed in clinical trials, a stronger decrease in actionability was observed in patients with lung cancer than in others, highlighting the high number of interventional studies based on gene alterations and involving these patients. The difference in actionability observed for metastatic disease could be due to an enrichment of lung cancer patients within this sub-population but could also reflect the different mutational profiles that have been observed in localized versus metastatic disease [32].

A genomic-directed treatment modification was declared in 17.3% of the patients. Considering the variety of tested samples in terms of tumor types, cancer staging, and lines of treatments, this result is consistent with previous findings using other CGP tests, and the two recent Japanese studies [19–26] using F1CDx test reporting 14% and 22.3%. In 44% of the cases, the test should most likely not have been ordered, because the patients had either a poor general condition or were in remission. However, due to the study design (implementation after free of charge program), it could be an opportunity for oncologists to perform a CGP test in some patients, and delay actionability until disease progression. This is probably the reason why stable disease was one of the main reasons for no treatment modification after the test. In addition, the impossibility of including patients in the proposed clinical trials was frequently reported as a reason for no-genomic-driven treatment. Notably, the CGP test was used to guide the therapeutic opportunities across the different lines of treatment. This relies on the assumption that the tumor genomic profile did not change over time and that the selected alterations were truncal ones, with stable addiction of tumor cells to the activated pathway. The clinical use of the proposed genomic-directed treatment over time might also, more pragmatically, reflects the exhaustion of standard of care possibilities, and a "last opportunity" to control the disease. In this context, new approaches are taking into account several actionable

molecular alterations to propose drug combinations for patients, in order to produce better survival outcomes [33].

This French study provides new information on the actionability of F1CDx for patients with solid tumors in oncology practice, and tends to confirm the interest of such CGP in guiding therapy in patients in a real-life setting. However, as this study was conducted on the basis of a FoC Program for F1CDx test, some oncologists had the opportunity to test tumor patients using a large panel content, including 1) those with stable disease at that time, and 2) those with previous genetic tests performed and effective ongoing matched drugs. Although these patients were included in the study, their treatment with a genomic-directed proposed therapy may not have been captured due to short follow-up, and if we add recall bias, under-evaluated.

The clinical impact of CGP may also be improved in the future due to interest and previous data in liquid biopsies which will allow testing patients for whom a tissue biopsy can't be realised and systematically at progression disease, as this sampling method is less invasive, safer, and faster than tissue biopsies [34–36].

## Conclusion

This French study provides new information on the actionability of the F1CDx test based on tissue samples, even though access to innovative matching molecules can still be limited. This trend confirms CGP's interest in clinical practice, particularly for patients with advanced or metastatic disease, and shows that the test results were used across the patient's lines of treatment, and sometimes months after their report to the clinician.

## Supporting information

**S1 Table. Clinical characteristics and F1CDx test results in metastatic patients.**
(DOCX)

**S2 Table. F1CDx Gene alterations (amplification, SNV, indel, rearrangement) in the evaluable population.**
(XLSX)

**S3 Table. Genomic directed treatments.**
(XLSX)

**S1 File. Clinical characteristics and F1CDx results in most frequent cancer types.**
(XLSX)

**S2 File. Genomic pathway groups altered in the evaluable population.** Occurrence of the 4 main classes of genomic alterations in genes belonging to 6 potentially targetable gene families/pathways (TK: tyrosine kinase, DDR: DNA damage repair, CC: cell cycle, PAM: PI3K-ATM-mTOR, RME: RAF-MEK-ERK, IE: immune evasion, OTH: others).
(DOCX)

## Acknowledgments

The authors would like to thank all patients who agreed to share their information for this project and the investigators who contributed to this project: Dr Kheir-Eddine Benmammar (Centre Hospitalier Emile Roux, Le-Puy-en-Velay); Dr Bruno Chauffert (CHU d'Amiens), Dr Stefano Chong Hun Kim (CHU de Besançon), Dr Elisabeth Luporsi (Hôpital de Mercy, CHR Metz-Thionville), Dr Thomas Walter (Hospices civils de Lyon), Dr Marie-Ange Mouret-Reynier (Centre Jean Perrin, Clermont-Ferrand), Dr Carolina Saldana (Hôpital Henri Mondor,

Créteil). We also thank IQVIA's support and Dr. Marie-Odile BARBAZA from AUXESIA (manuscript preparation and setup).

## Author Contributions

**Conceptualization:** Karen Leroy, Clarisse Audigier Valette, Alexandre Civet, Sandrine Galoin, Antoine Italiano.

**Formal analysis:** Karen Leroy, Clarisse Audigier Valette, Alexandre Civet, Sandrine Galoin, Antoine Italiano.

**Investigation:** Karen Leroy, Jérôme Alexandre, Lise Boussemart, Jean Chiesa, Clotilde Deldycke, Carlos Gomez-Rocca, Antoine Hollebecque, Jacqueline Lehmann-Che, Antoinette Lemoine, Sandrine Mansard, Jacques Medioni, Isabelle Monnet, Samia Mourah, Thomas Pierret, Dominique Spaëth.

**Methodology:** Alexandre Civet.

**Supervision:** Sandrine Galoin.

**Visualization:** Alexandre Civet, Sandrine Galoin.

**Writing – review & editing:** Karen Leroy, Alexandre Civet, Sandrine Galoin.

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
