## [Decision Letter · Decision Letter 0]

17 Apr 2023

PONE-D-22-35433Retrospective analysis of real-world data to evaluate actionability of a comprehensive molecular profiling panel in solid tumor tissue samples (REALM study)PLOS ONE

Dear Dr. Karen Leroy,

Thank you for submitting your manuscript to PLOS ONE. After careful consideration, we feel that it has merit but does not fully meet PLOS ONE’s publication criteria as it currently stands. Therefore, we invite you to submit a revised version of the manuscript that addresses the points raised during the review process.

ACADEMIC EDITOR:

At first, I wonder if the authors have provided the participants the chance of refusing this study by opening this study or not. We must obey not only the law but also medical ethics. This information is unclear in this manuscript.

Although the authors mentioned this study used real-world data, about a quarter of the participants had no metastatic lesions. The participants in this study do not represent real-world situation due to Free of charge for F1CDx.

However, this study may present the useful information to the clinicians after the revision depending on the reviewers' comments.

We look forward to receiving your revised manuscript.

Kind regards,

Tomoki Yamano, M.D. Ph.D.

Academic Editor

PLOS ONE

Journal Requirements:

this study was funded by Roche S.A.S

I have read the journal's policy and the authors of this manuscript have the following competing interests:

Karen Leroy: Roche- board, conference fees, scientific collaboration, scientific meeting fees; AstraZeneca, BMS- board, conference fees, scientific meeting fees; Lilly, Janssen- board ; Amgen, MSD, GSK- scientific meeting fees; Nanostring- conference fees, scientific collaboration.

Clarisse Audigier Valette: AstraZeneca, Boehringer Ingelheim- Financial Interests, Personal, Principal Investigator, Advisory Role; BMS, Lilly, Novartis, Pfizer-Financial Interests, Personal, Invited Speaker, Advisory Role; Abb Vie, GlaxoSmithKline, Janssen, MSD, Roche, Sanofi, Takeda-Financial Interests, Personal, Advisory Role. 

Antoine Italiano: Roche- Financial Interests, Personal, Advisory Board, Research Grant Roche. 

Reviewers' comments:

Reviewer's Responses to Questions

**Comments to the Author**

1. Is the manuscript technically sound, and do the data support the conclusions?

Reviewer #1: Yes

Reviewer #2: Yes

2. Has the statistical analysis been performed appropriately and rigorously? 

Reviewer #1: Yes

Reviewer #2: Yes

3. Have the authors made all data underlying the findings in their manuscript fully available?

Reviewer #1: No

Reviewer #2: Yes

4. Is the manuscript presented in an intelligible fashion and written in standard English?

Reviewer #1: Yes

Reviewer #2: Yes

5. Review Comments to the Author

Reviewer #1: The authors present the results of their genomic screening program using FoundationOne CDx (using both the genomic data and the report with proposed genomic-directed treatments). They performed a retrospective, non-interventional, multicentre study aimed to describe the molecular alterations in various cancers (mostly in lung cancers and rare cancers) and the cumulative incidence of the “actionability”. This is presented as a “routine” practice, which is questionable.

In terms of feasibility, the authors reach the same conclusions than what has already been published in terms of successful molecular analyses and low rate (17.6%) of patients actually treated with matched therapy. However, the authors do not present any outcome parameters. Why are response rate data not reported (at least for lung cancer patients)?

It is not clear what was the proportion of patients who received matched therapy available within the indication (such as EGFR inh. in EGFR-mutated NSCLC) versus outside their indication. FoundationOne CDx used in this program would not have changed anything in patients treated within the indications (especially for lung cancer patients since looking for these alterations is standard). The authors should describe that.

With regard to the recommendations given, did the recommendations vary in their strength? The authors did mention in the Discussion that using the ESCAT (ESMO Scale for Clinical Actionability of molecular Targets) would have been noteworthy, but it is difficult to understand why “insufficient information on gene alterations was collected to re-analyse actionability”?

The authors should also address the question of cost-benefit of this real-world retrospective French program. What would have been the overall costs if the FoundationOne CDx had been paid for? Can this be commented on?

Reviewer #2: This is a real world study of the ability of a NGS panel performed in cancer patients in terms of actionability and modification of treatments.

The definition of actionability must be more detailed. The text and Fig 2 must reflect the 3 possible categories individually in order to offering a better applicability of the observations (approved and financed in individual case, approved in other cancer types, and only available in trials)

There must be a detailed account of why a Foundation Medicine test was ordered in patients without metastatic disease (25% of the cases), since any test result in this setting will most likely not have an impact on treatment. Also, patients with non-metastatic disease should be analyzed separately from the metastatic cases.

Table 1 should have a column for cholangiocarcinomas and include rare tumors with the "other tumors" category

For the actionability section of results, the authors must express the "actual differential actionability gain", i.e. how many cases received a genomic treatment because of F1CDx alone (and in how many cases standard molecular sytudies would have been enough)

The authors must explain clearly in the results section and in the discussion that in 44% of the cases (23%+20.9%) the test should most likely not have been ordered, because the patients had either a poor general condition of were in remission (here they must clarify if these last patient group are the patients with non-metastatic disease)

References do not correspond in many instances with the text and must be revised completely

6. PLOS authors have the option to publish the peer review history of their article (what does this mean?). If published, this will include your full peer review and any attached files.

Reviewer #1: No

Reviewer #2: No

---

## [Author Response · Author response to Decision Letter 0]

30 May 2023

Dear Editor, dear reviewers

We thank you for considering our manuscript entitled «Retrospective analysis of real-world data to evaluate actionability of a comprehensive molecular profiling panel in solid tumor tissue samples (REALM study)» for publication in PLOS One.

We have read carefully the comments and requests of the academic editor and reviewers. We have answered these comments and have modified the manuscript accordingly, as explained in detail below.

Academic editor comment “I wonder if the authors have provided the participants the chance of refusing this study by opening this study or not”.

As specified in the specific comment upon initial submission, the patients were informed at the time of the tumor testing (F1CDx) and gave a written consent to the test. They were subsequently informed of the proposed retrospective study before the beginning of fully anonymized data collection. They had the possibility to oppose to the use of their data and the eCRF could only be filled if the clinician certified that the patient did not oppose to this study. The manuscript has been modified page 5 to clarify this point.

Reviewer 1.

“However, the authors do not present any outcome parameters. Why are response rate data not reported (at least for lung cancer patients)?”

We did not specifically collect the clinical outcome after treatment change (ie response to therapy), we only collected the date of treatment start and stop, and not the reason why the treatment was stopped. Unfortunately, these data are not sufficient to evaluate the outcome under genomic-driven therapy. We have changed the methods section page 6 which incorrectly mentioned “disease evolution” that may have mislead the reviewer and let him think that we did not make all data underlying our findings available, as follows “after F1CDx report availability in the center (changes in patient treatment, according to the test results or not, and reasons for no changes, date of treatment start and stop, date of last news”.

“It is not clear what was the proportion of patients who received matched therapy available within the indication (such as EGFR inh. in EGFR-mutated NSCLC) versus outside their indication”.

The REALM study was observational and based on a Free of Charge program that was used differently in the different centers: as a standard diagnosis test in some of them, as a complementary test to search for alterations which could allow inclusion in clinical trial, or as an exploratory test in rare diseases. According to the data showed in S3 Table, 72 pts had a treatment change based on F1CDx results declared by the clinician, and 49 of these were not standard care at that time. Most lung cancer and melanoma patients did receive matched therapy within the indication, presumably because the test was ordered at diagnosis instead of or in addition to standard testing. We have added this information in the revised version of S3 Table and on page 12 of the manuscript “This genomic-directed treatment could be standard of care in the indication (23 pts), investigational drugs within clinical trials (20 pts) or off label use of an available molecule (29 cases)”. 

Details : 1 pt with colo-rectal carcinoma (01-21) received larotrectinib after identification of a NTRK3 fusion; 12 lung cancer pt (04-01, 04-02, 04-06, 04-33, 04-34, 04-51, 14-01, 14-013, 14-04, 04-03, 04-07, 07-04) received TKI, chemotherapy or BRAF inhibitors that were standard therapies and 8 lung cancer pts received treatment that were investigation drugs at that time (EGFR, MET, RET and KRAS inhibitors), 10 melanoma pts (08-01, 08-14, 13-01, 13-05 and 13-13, 08-07, 08-07, 08-15, 13-10, 13-11) received ICI, BRAF or MEK inhibitors that were standard of care, whereas 2 melanoma pts received other drugs. 

“Did the recommendations vary in their strength? The authors did mention in the Discussion that using the ESCAT (ESMO Scale for Clinical Actionability of molecular Targets) would have been noteworthy, but it is difficult to understand why “insufficient information on gene alterations was collected to re-analyse actionability”?

The F1CDx report provides the information whether the treatment is an approved drug in patient’s tumor type or in another tumor type, or investigational drug in a clinical trial. Approved drugs are based on FDA authorizations, which in fact does not apply to France (ie high TMB is not recognized as an indication for anti-PD1/PD-L1 treatment in France). Therefore, the recommendation may have changed between April 2017 and September 2019. Regarding ESCAT, we collected the name of the gene, and type of genetic alteration (ie SNV, indel, amplification or fusion), but not the precise change, and this does not allow for ESCAT assessment for SNV/indel variants (which will be different according to the position and change on the gene and protein).

“The authors should also address the question of cost-benefit of this real-world retrospective French program. What would have been the overall costs if the FoundationOne CDx had been paid for?”

This question cannot be really addressed because of 1) the heterogeneity in test ordering (standard diagnosis tests, complementary tests for clinical trial, exploratory tests…); 2) “regular” biomarker testing costs are also heterogeneous, depending on the technology used (NGS or targeted testing); 3) evaluation of cost-benefit should also include the impact on medical care costs, which was far beyond the scope of this study.

Reviewer 2

“The definition of actionability must be more detailed. The text and Fig 2 must reflect the 3 possible categories individually in order to offering a better applicability of the observations (approved and financed in individual case, approved in other cancer types, and only available in trials)”

In the F1CDx report, there were several alterations identified for each tumor, and for each of these, there was an indication whether it was associated with approved therapy in the tumor type/ approved in other cancer type / available in clinical trial (wherever in the world). In order to answer the reviewer’s comment, we have added the information requested, using the highest recommendation level for each tumor in the revised Fig 2 and in the text page 12 “In the evaluable population of 416 patients, F1CDx test actionability was 75.0%, including drugs approved in the cancer type (40.7%), in another cancer type (42.3%), or only available in trials (17.0%).”

“There must be a detailed account of why a Foundation Medicine test was ordered in patients without metastatic disease (25% of the cases), since any test result in this setting will most likely not have an impact on treatment. Also, patients with non-metastatic disease should be analyzed separately from the metastatic cases.”

As previously mentioned, clinicians were free to select the patients for which would order the F1CDx test, and some of them decided to test non-metastatic pts, presumably because they had rare disease and/or expected a relapse and requirement for precision medicine at some point of the patient evolution. Although we acknowledge the fact that in this setting, testing will most likely have no impact on treatment, we have uncovered that testing results were indeed used to propose genomic-oriented treatment months after testing (see Fig 4 and Sup Table 3) and we think this is an important information that we discussed in page 16-17. In order to answer to the reviewer comment, we have modified the text page 8 and 12, modified the Figure 2 (see previous point), and added a S1 Table showing the clinical characteristics and results of the F1CDx test corresponding to the data of Table 1 and Table 2, but only in patients with metastatic disease.

“Table 1 should have a column for cholangiocarcinomas and include rare tumors with the "other tumors" category”

Only lung cancer and rare tumors were individualized in the Tables 1 and 2 presented in the manuscript because of their important representation in this series (lung cancer) or high level of heterogeneity (rare tumors). To answer this specific request, we have modified the text page 8 and added a S1 File to show the clinical characteristics and results of the F1CDx test, in all evaluable patients and in metastatic disease, according to 5 different cancer types with more than 20 cases (biliary tract, skin, brain, bladder and breast). 

“For the actionability section of results, the authors must express the "actual differential actionability gain", i.e. how many cases received a genomic treatment because of F1CDx alone (and in how many cases standard molecular studies would have been enough)”

Please see response to commentary 2, reviewer 1. Cases for which standard molecular studies would have been enough are the 23/72 pts with a genomic-directed treatment considered as standard of care in the indication. Therefore, the “actual differential actionability gain” is estimated to be 68% (49/72) in pts who received a genomic directed treatment. However, given the declarative and descriptive characteristic of this study, this percentage has to be considered with caution and is not included in the main text of the revised manuscript.

“The authors must explain clearly in the results section and in the discussion that in 44% of the cases (23%+20.9%) the test should most likely not have been ordered, because the patients had either a poor general condition of were in remission (here they must clarify if these last patient group are the patients with non-metastatic disease)”

We agree with the reviewer comment. This information was clearly stated in the result section (page 12) and we have added this sentence in the discussion page 16. The situation for metastatic patients has been clarified in the revised Fig 2, and supplementary Table 1 and File 1 (see above).

“References do not correspond in many instances with the text and must be revised completely”

We have checked all references and corrected all errors.

We hope that you will consider this revised version of our manuscript is suitable for publication in your journal.

Best regards, Pr Karen Leroy

---

## [Decision Letter · Decision Letter 1]

31 Aug 2023

Retrospective analysis of real-world data to evaluate actionability of a comprehensive molecular profiling panel in solid tumor tissue samples (REALM study)

PONE-D-22-35433R1

Dear Dr. Leroy,

We’re pleased to inform you that your manuscript has been judged scientifically suitable for publication and will be formally accepted for publication once it meets all outstanding technical requirements.

Kind regards,

Tomoki Yamano, M.D. Ph.D.

Academic Editor

PLOS ONE

Additional Editor Comments (optional):

The authors revised the manuscript and the reviewer 1 and reviewer 3 agreed their response was appropriate for the reviewers comments from reviewer 1 and reviewer 2. However, reviewer 3 presented the additional comments regarding the timing of NGS and small number of the assessed cases. Reviewer 1 and reviewer 2 and I also recognize these problems which are not revised any more. I believe the readers may obtain some knowledge from this manuscript in spite of these defects, especially from the data from variety of cancer types.

Reviewers' comments:

Reviewer's Responses to Questions

**Comments to the Author**

1. If the authors have adequately addressed your comments raised in a previous round of review and you feel that this manuscript is now acceptable for publication, you may indicate that here to bypass the “Comments to the Author” section, enter your conflict of interest statement in the “Confidential to Editor” section, and submit your "Accept" recommendation.

Reviewer #2: All comments have been addressed

Reviewer #3: All comments have been addressed

2. Is the manuscript technically sound, and do the data support the conclusions?

Reviewer #2: Yes

Reviewer #3: Yes

3. Has the statistical analysis been performed appropriately and rigorously? 

Reviewer #2: Yes

Reviewer #3: Yes

4. Have the authors made all data underlying the findings in their manuscript fully available?

Reviewer #2: Yes

Reviewer #3: Yes

5. Is the manuscript presented in an intelligible fashion and written in standard English?

Reviewer #2: Yes

Reviewer #3: No

6. Review Comments to the Author

Reviewer #2: Thanks for addressing all points that were raised in the first version of the manuscript. The current revision is fine

Reviewer #3: Dear Author,

Thank you for the opportunity to review the manuscript of "Retrospective analysis f real-world data to evaluate actionability of a comprehensive molecular profiling panel in solid tumor tissue samples (REALM study)".

After reviewing the replies to the previous reviewers, all the comments have been addressed. My only comments is: Patients who had NGS done 6-24 months prior were again more likely to use the NGS results to guide therapy. What were the characteristics of these patients? Were they different from those who use the NGS results for matching in the first 6 months of NGS results?

Aside from that, some English editing is needed?

7. PLOS authors have the option to publish the peer review history of their article (what does this mean?). If published, this will include your full peer review and any attached files.

Reviewer #2: No

Reviewer #3: No

---

## [Editor Report · Acceptance letter]

5 Sep 2023

PONE-D-22-35433R1 

Retrospective analysis of real-world data to evaluate actionability of a comprehensive molecular profiling panel in solid tumor tissue samples (REALM study) 

Dear Dr. Leroy:

I'm pleased to inform you that your manuscript has been deemed suitable for publication in PLOS ONE. Congratulations! Your manuscript is now with our production department. 

Kind regards, 

on behalf of

Dr. Tomoki Yamano 

Academic Editor

PLOS ONE